# Respiratory Physiology of COVID-19 and Influenza Associated Acute Respiratory Distress Syndrome

**DOI:** 10.3390/jcm11216237

**Published:** 2022-10-22

**Authors:** Niklas Kronibus, Frederik Seiler, Guy Danziger, Ralf M. Muellenbach, Christian Reyher, André P. Becker, Maren Kamphorst, Torben M. Rixecker, Carlos Metz, Robert Bals, Philipp M. Lepper, Sebastian Mang

**Affiliations:** 1Interdisciplinary COVID-19-Center, University Medical Centre, Saarland University, 66421 Homburg, Germany; 2Department of Internal Medicine V–Pneumology, Allergology, Critical Care and ECMO/ECLS Center Saar, University Medical Centre, Saarland University, 66421 Homburg, Germany; 3Department of Anesthesiology and Critical Care Medicine, Campus Kassel of the University of Southampton, 34125 Kassel, Germany

**Keywords:** coronavirus-disease 2019 (COVID-19), acute respiratory distress syndrome (ARDS), mechanical ventilation, influenza A/B, extracorporeal membrane oxygenation (ECMO)

## Abstract

**Background:** There is ongoing debate whether lung physiology of COVID-19-associated acute respiratory distress syndrome (ARDS) differs from ARDS of other origin. **Objective**: The aim of this study was to analyze and compare how critically ill patients with COVID-19 and Influenza A or B were ventilated in our tertiary care center with or without extracorporeal membrane oxygenation (ECMO). We ask if acute lung failure due to COVID-19 requires different intensive care management compared to conventional ARDS. **Methods**: 25 patients with COVID-19-associated ARDS were matched to a cohort of 25 Influenza patients treated in our center from 2011 to 2021. Subgroup analysis addressed whether patients on ECMO received different mechanical ventilation than patients without extracorporeal support. **Results**: Compared to Influenza-associated ARDS, COVID-19 patients had higher ventilatory system compliance (40.7 mL/mbar [31.8–46.7 mL/mbar] vs. 31.4 mL/mbar [13.7–42.8 mL/mbar], *p* = 0.198), higher ventilatory ratio (1.57 [1.31–1.84] vs. 0.91 [0.44–1.38], *p* = 0.006) and higher minute ventilation at the time of intubation (mean minute ventilation 10.7 L/min [7.2–12.2 L/min] for COVID-19 vs. 6.0 L/min [2.5–10.1 L/min] for Influenza, *p* = 0.013). There were no measurable differences in P/F ratio, positive end-expiratory pressure (PEEP) and driving pressures (ΔP). Respiratory system compliance deteriorated considerably in COVID-19 patients on ECMO during 2 weeks of mechanical ventilation (C_rs_, mean decrease over 2 weeks −23.87 mL/mbar ± 32.94 mL/mbar, *p* = 0.037) but not in ventilated Influenza patients on ECMO and less so in ventilated COVID-19 patients without ECMO. For COVID-19 patients, low driving pressures on ECMO were strongly correlated to a decline in compliance after 2 weeks (Pearson’s R 0.80, *p* = 0.058). Overall mortality was insignificantly lower for COVID-19 patients compared to Influenza patients (40% vs. 48%, *p* = 0.31). Outcome was insignificantly worse for patients requiring veno-venous ECMO in both groups (50% mortality for COVID-19 on ECMO vs. 27% without ECMO, *p* = 0.30/56% vs. 34% mortality for Influenza A/B with and without ECMO, *p* = 0.31). **Conclusion:** The pathophysiology of early COVID-19-associated ARDS differs from Influenza-associated acute lung failure by sustained respiratory mechanics during the early phase of ventilation. We question whether intubated COVID-19 patients on ECMO benefit from extremely low driving pressures, as this appears to accelerate derecruitment and consecutive loss of ventilatory system compliance.

## 1. Background

The emergence of COVID-19 in December 2019 brought a surge of patients with viral pneumonia and associated acute respiratory distress syndrome (ARDS) [1,2].

While dexamethasone [3], tocilizumab [4] and the availability of Anti-COVID-19 vaccines have improved survival in some patients, mortality of COVID-19-associated ARDS (CARDS) on intensive care units still reaches 50% and depends heavily on invasive mechanical ventilation (IMV) and veno-venous extracorporeal membrane oxygenation [5,6]. Although live-saving in many cases, IMV may also contribute to lung damage through ventilator-associated lung injury (VILI). We performed a single-center retrospective analysis to compare lung physiology and mechanical ventilation settings in patients with CARDS and Influenza-associated ARDS. Our goal was to elucidate whether optimal mechanical ventilation strategies for CARDS patients differ from those applied in ARDS of other origin.

Shortly into the first pandemic wave, observations arose that respiratory failure due to COVID-19 could have unusual physiologic features [7,8,9,10]. In May 2020, Marini et al. described patients with COVID-19 as often presenting with severe hypoxemia yet only mildly impaired respiratory mechanics [11]. Since ARDS was historically accompanied by a loss of functional residual capacity and compliance [12], this observation sparked a debate whether respiratory failure due to COVID-19 might be a different disease. To this day, this question has not yet been sufficiently answered. Another recurrent observation in COVID-19 is a disproportionate increase in respiratory effort [13], visible through elevated minute ventilation, leading to only marginal clearance of carbon dioxide (CO_2_). Several mechanisms have been suggested to account for this discrepancy, e.g., ventilation-perfusion mismatch with increased functional right-to-left shunting [14], increased CO_2_ production [15] or a combination of both. The question of whether COVID-19-associated ARDS is different from ‘classic’ ARDS is not only of academic purpose, as it might imply a need for different intensive care management [8,16,17].

Early in the first COVID-19 pandemic wave, centers started to employ extracorporeal membrane oxygenation for the most severe COVID-19 cases, drawing on existing cohort data on ECMO for ARDS of different origin [18,19]. While ECMO has proven to reduce mortality of patients with very severe COVID-19 [5], the mortality benefit for these patients was unsatisfyingly low [20]. It is well-accepted that high-driving pressures may increase mortality of ARDS patients through ventilator-associated lung injury (VILI) [21]. Consequently, these patients usually receive high positive end-expiratory pressure (PEEP) and low driving pressures at the cost of CO_2_ elimination to limit mechanical power on the lungs [22]. The strength of ECMO for ARDS patients is considered not only providing oxygenation but allowing protective ventilation through effective extracorporeal CO_2_ clearance. If ventilator-induced lung injury contributes to disease progression in COVID-19, then protective ventilation during ECMO should decelerate the deterioration of lung mechanics.

## 2. Methods

### 2.1. Study Design

Our ICU is a tertiary care unit with 24 beds dedicated to the treatment of COVID-19 patients. Most COVID-19 patients included in the study were intubated in primary or secondary care centers nearby and then transferred for evaluation of ECMO. Several patients were cannulated externally by our mobile ECMO teams and then transferred to Homburg on running ECMO. We first analyzed basic characteristics of 25 patients with COVID-19, treated in our center from March 2020 to March 2021. All included patients were mechanically ventilated and devoid of severe comorbidities. Patients aged between 18 and 70 with laboratory-confirmed SARS-CoV-2 infection, defined as a positive result on real-time RT-PCR assay from nasal or pharyngeal swabs or respiratory tract aspirates, with hypoxemia and a high respiratory drive meeting the criteria for moderate ARDS according to the Berlin definition of ARDS were included in this cohort. Ethical approval was waived by the institutional review board (IRB), which is the Ärztekammer des Saarlandes. Informed consent for the analysis of data is waived by the IRB (Ärztekammer des Saarlandes) due to the anonymous and retrospective analysis of data. Data collection and methods used in this manuscript were carried out in accordance with relevant guidelines and regulations, in particular the Saarländisches Krankenhausgesetz (Law for the regulation of research conducted in hospitals of the federal state of Saarland) [23].

Most COVID-19 patients treated in our center could not be recruited for the study cohort due to one of the following exclusion criteria: age- or comorbidity-related contraindication for extracorporeal life support, additional circulatory failure, defined as requiring intravenous noradrenaline of more than 0.3 µg/kg/min, cardiopulmonary resuscitation prior to intubation for COVID-19-associated ARDS, mechanical ventilation for COVID-19 for more than 8 days prior to study inclusion.

14 patients from the COVID-19 cohort later received veno-venous extracorporeal membrane oxygenation due to respiratory failure.

A matched control cohort of 25 patients with ARDS due to Influenza A or B, treated in our center from 2011 to 2021, was then recruited for comparison. As severe COVID-19 affects predominantly men, matching both cohorts for sex was not possible. Instead, both groups were matched according to age and BMI as well as possible. The same exclusion criteria were applied for the recruitment of Influenza patients.

We compared the following parameters taken from day 1, 2, 4, 8 and 16 of invasive ventilation: positive end-expiratory pressure (PEEP); plateau pressure (PP); driving pressure (ΔP); and Respiratory Rate (RR).

Ventilation was in pressure-controlled mode in all patients. Generally, we targeted a protective ventilation approach, aiming at driving pressures below 15 cmH_2_O and tidal volumes (VT) of approximately 6 mL per kg of predicted bodyweight (PBW). The mean arterial pressure target was 60–65 mmHg if physiological aims were reached. These included capillary refill time (i.e., warm periphery), sufficient urinary output (≥0.5 mL/kg/h) and normal lactate levels (≤2.0 mmol/L). Norepinephrine was the vasopressor of choice in patients with pH ≥ 7.25. Inotrope use was initiated if ScvO_2_ was below 65% despite adequate hemoglobin levels. Nutrition in both groups was done according to the same standards.

If patients were not deeply sedated and breathed spontaneously, the actual rate of assisted spontaneous breathing was counted whenever intended, and actual respiratory rates diverged by more than 2 per minute.

Derived from the above-mentioned parameters as well as results from blood gas samples, we calculated the following parameters: P_a_O_2_/F_i_O_2_ ratio (arterial oxygen partial pressure divided by inspiratory oxygen fraction); ventilatory system compliance, both absolute and in relation to idealized body weight (tidal volume in relation to predicted body weight divided by driving pressure); and ventilatory ratio (VR) as a measure of ventilator efficiency, calculated as (VE_real_ × PaCO_2real_) divided by (100 mL/min × predicted body weight (PBW; kg) × 40 mmHg (expected PaCO_2_)).

We defined the primary endpoint “death from any cause” to calculate 28-day and 60-day mortality for both cohorts and subgroups as well as the secondary endpoint “free from ventilator support after 15 days of ventilation” to compare weaning success in both groups.

### 2.2. Statistical Analysis

Continuous variables are displayed as mean, standard deviation and 95% confidence intervals. For ordinal variables and non-normally distributed continuous variables, we calculated median and interquartile range.

Horizontal comparisons between groups were done by two-sided t-testing for normally distributed continuous variables and the Mann–Whitney U test for non-continuous parameters. Changes in parameters over time were analyzed by paired two-sided t-testing at a significance level of 0.05. Kaplan–Meier curves were plotted using the corresponding SPSS tool with “death from any cause” as counted event. We performed log rank tests to compare survival between groups. Analysis and plotting were performed with SPSS Statistics v. 2.6.0.0 by IBM, Armonk, New York, United States of America.

## 3. Results

### 3.1. Basic Patient Characteristics

Basic patient characteristics are displayed in Table 1. As severe COVID-19 affects predominantly men, the COVID-19 group has an excess of male patients (20 of 25 (80%)) vs. 16 of 25 (64%), *p* = 0.21.

Both groups showed severe hypoxemia at the time of intubation. Mean arterial oxygen partial pressure before intubation was slightly higher in the influenza group (COVID19: paO2 84.4 mmHg [76.8–92.1 mmHg] vs. 93.3 mmHg [71.8–114.8 mmHg] for Influenza, *p* = 0.31). Prior to intubation, patients received either oxygen supplementation via high-flow nasal cannula or non-invasive ventilation. Mean inspiratory oxygen fraction during non-invasive oxygenation was slightly lower in the COVID-19 group (0.6 [0.49–0.70], n = 14) compared to control (0.75 [0.53–0.84], n = 9).

Thirteen of 25 COVID-19 patients (42%) and 16 of 25 Influenza (64%) patients received veno-venous extracorporeal membrane oxygenation for respiratory failure a few days after intubation.

### 3.2. Horizontal Comparison of Ventilation Parameters

On the first day of invasive ventilation, there were no significant differences between both groups in P/F ratio (127.9 mmHg [112.8–161.3 mmHg] for COVID-19 vs. 135.4 mmHg [100.3–180.4] mmHg, *p* = 0.56); positive end-expiratory pressure (12.0 mbar [9.6–13.5 mbar] for COVID-19 vs. 13.0 mbar [9.0–15.5 mbar] for Influenza, *p* = 0.37); plateau pressure (23.5 mbar [20.8–29.0 mbar] for COVID-19 vs. 25.0 mbar [24.0–28.5 mbar] for Influenza, *p* = 0.56); or driving pressure (13.5 mbar [10.8–16.0 mbar] for COVID-19 vs. 12.0 mbar [11.0–14.5 mbar] for Influenza, *p* = 0.64).

Similar ventilation pressures produced higher tidal volumes in relation to predicted body weight in COVID-19 patients (7.69 mL/kgBW [7.12–8.12 mL/kgBW]) compared to control (5.12 mL/kgBW [3.14–8.40 mL/kgBW], *p* = 0.06), indicating less impaired ventilatory system compliance in the COVID-19 group. COVID-19 patients required significantly higher ventilation frequencies (20.0 min^−1^ [15.5–21.3 min^−1^] vs. 14.0 min^−1^ [12.5–17.5 min^−1^], *p* = 0.011) and minute ventilation (10.7 L/min [7.2–12.2 L/min] vs. 6.0 L/min [2.5–10.1 L/min], *p* = 0.013) for sufficient CO_2_ elimination. Indeed, calculated static compliance of the respiratory system was higher in COVID-19 patients than in Influenza patients (40.7 mL/mbar [31.8–46.7 mL/mbar] vs. 31.4 mL/mbar [13.7–42.8 mL/mbar], *p* = 0.198) throughout the observation period. We calculated ventilatory ratio (VR) for both groups to assess ventilation efficacy. VR was significantly higher in the COVID-19 cohort (1.57 [1.31–1.84] vs. 0.91 [0.44–1.38], *p* = 0.006). For a comprehensive overview of ventilation parameters, see Table 2 and Table 3 and Figure 1.

### 3.3. Longitudinal Comparison of Ventilation Parameters

Immediately after intubation, most patients in both groups received pressure-controlled ventilation (PCV) during deep sedation or relaxation. The proportion of patients with strictly controlled ventilation on the first day was 79% in the COVID-19 group (11/14) and 100% in the Influenza group (9/9). All patients for whom ventilation data from the first day of mechanical ventilation were missing could not be included in the analysis.

As ventilation progressed, an increasing number of patients could be weaned to augmented spontaneous breathing (ASB) with continuous positive airway pressure (CPAP). On day 8, 41% of COVID-19 patients and 38% of Influenza patients breathed spontaneously with pressure support. On day 16, the rate of patients still on controlled ventilation had dropped to 9.5% in the COVID-19 group (2/21) but stayed relatively unchanged at 36% (4/11) in the control group.

Except for a few Influenza patients developing prolonged disease, we observed that COVID-19 patients were ventilated much longer than most Influenza patients. Odds Ratio in the overall cohort for breathing free from ventilator 15 days after intubation was 3.5 for Influenza compared to COVID-19.

To assess how ventilatory system compliance developed over time, we performed longitudinal pair-wise t-test comparisons of the initial compliance from the day of intubation to day 8 and day 16 of invasive ventilation.

We detected a continuous decrease of compliance in the COVID-19 cohort, with a significant reduction after 2 weeks of ventilation (47.99 mL/mbar ± 32.80 mL/mbar vs. 24.13 mL/mbar ± 10.70 mL/mbar, mean difference after 15 days being −23.87 mL/mbar ± 32.94 mL/mbar, *p* = 0.037) (Figure 2).

In contrast, ventilatory system compliance improved in the Influenza cohort after 7 days (28.32 mL/mbar ± 21.98 mL/mbar vs. 34.85 mL/mbar ± 29.59 mL/mbar, *p* = 0.10). Since the course of ARDS tended to be much shorter for Influenza patients, we cannot provide a comparison in this cohort over 15 days, as only 2 patients would have qualified for pair-wise comparison between days 1 and 16.

### 3.4. Subgroup Analysis–ECMO vs. Non-ECMO

Gradual decline in compliance was particularly present in the subgroup of COVID-19 patients receiving extracorporeal support (mean loss of compliance after 7 days 19.5 mL/mbar ± 42.6 mL/mbar, *p* = 0.31, mean loss after 15 days: −36.19 mL/mbar ± 41.88 mL/mbar, *p* = 0.09). COVID-19 patients not on ECMO also deteriorated in compliance; however, the loss of compliance was not as strong (−9.08 mL/mbar ± 4.43 mL/mbar after 15 days, *p* = 0.010).

After ECMO was initiated in most patients, we checked whether ventilation strategies changed following ECMO initiation. Indeed, for the COVID-19 group, patients on ECMO received significantly higher PEEP on days 8 and 16 compared to patients without ECMO (Day 8: 12.5 mbar ± 2.5 mbar vs. 10.3 mbar ± 2.0 mbar, *p* = 0.025, Day 16: 12.3 mbar ± 2.3 vs. 7.8 mbar ± 1.8 mbar, *p* < 0.001). Attempting to limit VILI, patients on ECMO were ventilated with lower driving pressures. This difference was small for day 8 as not all patients had been cannulated up to that day, but the difference was significant for day 16 (Day 8: 13.8 mbar ± 4.6 vs. 14.2 mbar ± 2.6 mbar, *p* = 0.83, Day 16: 11.3 mbar ± 2.3 mbar vs. 16.6 mbar ± 4.5 mbar, *p* = 0.002).

In contrast, there were no differences between PEEP and driving pressures between patients with and without ECMO for the Influenza group (not shown).

Since patients on ECMO tended to receive lower driving pressures but also seemed to deteriorate in ventilatory system compliance over time, we checked for correlations between a reduction in driving pressure and decreases in compliance over 15 days. We found that decreased driving pressures, as a strategy of protective ventilation on ECMO, was positively correlated to a loss of compliance after 15 days in bivariate correlation analysis (Pearson’s R = 0.80, *p* = 0.058) in the COVID-19 group. In contrast, COVID-19 patients not receiving ECMO showed an inverse correlation between change in driving pressure and compliance on day 16 (Pearson’s R = -0.92, *p* = 0.025). Compared to the COVID-19 group, this correlation was weak for Influenza patients on ECMO between the first and eighth day of ventilation (Pearson-coefficient −0.10, *p* = 0.88).

### 3.5. Outcome

We determined “death from any cause” as an endpoint to assess survival rates after 28 and 60 days from the beginning of invasive ventilation.

Twenty-eight-day mortality was 16% for the COVID-19 group compared to 36% in the control group. As Kaplan–Meier plots demonstrate (Figure 3A–C), overall survival was slightly better for the COVID-19 group compared to control (40% vs. 48% for Influenza, *p* = 0.31). Requiring veno-venous ECMO was a negative predictive factor for survival in both groups (50% mortality for COVID-19 on ECMO vs. 27% without ECMO, *p* = 0.30/56% vs. 34% mortality for Influenza A/B with and without ECMO, *p* = 0.31).

## 4. Discussion

Our study provides evidence that, during the early phase of mechanical ventilation, COVID-19-associated ARDS differs from Influenza-associated ARDS, mainly in lung mechanics. The ventilator ratio in patients with COVID-19-associated ARDS is statistically significant higher than in patients with Influenza-associated ARDS. Thus, the efficacy of CO_2_ elimination in COVID-19-associated ARDS is highly impaired.

The finding that COVID-19 patients present with higher minute ventilation to achieve similar CO_2_ clearance is in line with results from other groups [10,24]. Recently, we were able to demonstrate that pulmonary shunt fraction is not elevated in COVID-19 patients compared to an ARDS cohort [15]. Increased ventilatory effort is the result of heavily increased CO_2_ production rather than right-to-left shunting. Almost all COVID-19 patients had high fevers; some as high as 40 °C, which might have contributed to CO_2_ accumulation. A recent retrospective analysis has connected fever of COVID-19 patients on admission to worse outcome [25]. Aggressive medical treatment of hyperthermia could limit respiratory drive and improve CO_2_ clearance in both awake and intubated patients.

Although simplifying a complex disease, the concept of L-and H-type COVID-19 introduced by Gattinoni et al. [26] reflects the observation that patients with COVID-19 tend to transition from sustained to impaired lung mechanics.

Early into the pandemic, intubation of COVID-19 patients was recommended permissively, mainly to prevent emergency intubation during acute deterioration and to limit aerosol generation during non-invasive ventilation or high-flow oxygen supplementation [27]. This approach has been questioned by emerging insight into COVID-19 lung physiology [28]. The main advantages of invasive ventilation are maintenance of high positive end-expiratory pressure and facilitating lung recruitment. If patients with early COVID-19-associated respiratory failure possess sustained lung mechanics, it appears questionable whether they benefit from early intubation, as derecruitment is probably not the leading cause of hypoxemia. This insight has raised the dilemma about which COVID-19 patients should be mechanically ventilated, provided they do not require emergency intubation, and if mechanically ventilated, when the intubation should be performed.

Awake patients with COVID-19 pneumonia typically produce high tidal volumes with high negative intrathoracic pressures during non-invasive ventilation or high-flow nasal cannula [29]. The resulting mechanical strain puts them at risk of developing patient self-inflicted lung injury (pSILI), a concept which has already been described for other conditions with respiratory failure [30]. On the other hand, once patients are intubated, the need for CO_2_ control will most likely limit the possibilities of ventilating patients with tidal volumes low enough to limit ventilator-associated lung injury (VILI). Indeed, those COVID-19 patients from our cohort who did not receive veno-venous ECMO required higher tidal volumes and minute ventilation than what most physicians would consider ‘protective’ for ARDS. Furthermore, invasive ventilation increases the risk of ventilator-associated infections, possibly accelerating the transition from L- to H-type COVID-19 [26]. This assumption is in line with our data which show that intubated COVID-19 patients without ECMO required relatively high driving pressures throughout the course of disease. Interestingly, it is the ECMO group of COVID-19 patients in our cohort who had the heaviest losses in respiratory system compliance, possibly and paradoxically due to low driving pressures.

Our data suggest that reducing ventilation effort to a more protective scheme (e.g., 3–5 mL/kg predicted body weight) in patients on ECMO might give way to a subsequent loss of respiratory system compliance. We assume that reduced alveolar gas exchange might lead to further alveolar derecruitment in these patients. Subsequent alveolar edema might finally lead to the condition that Gattinoni and others have described as ‘H-type’ COVID-19 [26]. This effect was present even though, as outlined above, COVID-19 patients on ECMO had higher average PEEP than those without ECMO, suggesting that an increase in PEEP alone is not sufficient to prevent derecruitment in these patients. While it is true that the danger of fatal hypoxia will probably be averted by ECMO, it is unknown whether this advantage can outweigh the long-term complications associated with derecruitment.

We did not detect a loss of compliance in our Influenza patients on ECMO, which is probably because respiratory system compliance was already severely impaired in these patients when they were intubated, hence protective ventilation on ECMO did not further jeopardize recruitment. As our mortality data shows, many COVID-19 patients do not recover from this state, questioning the role of highly protective ventilation on ECMO for COVID-19. Theoretically, initiating awake ECMO in COVID-19 patients could provide sufficient gas exchange without the need for invasive mechanical ventilation. First cohort studies on an awake ECMO approach for COVID-19 have shown discouraging results, possibly because awake patients are at greater risk of patient self-inflicted lung injury [31]. Future studies may reveal whether alveolar derecruitment, following vvECMO initiation due to ultraprotective ventilation, outweighs lung damage caused by less protective ventilation strategies.

## 5. Limitations

Our study has several limitations that need to be addressed. First, as we present real-life data, the matching of COVID-19 and control cohorts was not ideal, since COVID-19 patients were noticeably, yet not significantly, older compared to control. In addition, the COVID-19 group had a significant predominance of male patients, and some comorbidities were unevenly distributed among both groups. Secondly, being a tertiary care center, many patients were intubated in external hospitals and later transferred to our center for ECMO evaluation. Hence, data on the first days of ventilation is missing for some patients. As this is the case for both COVID-19 and Influenza patients, systematic error will probably be limited. Finally, given that COVID-19 is a relatively new disease and Influenza patients were recruited from the last 10 years, we cannot exclude that changes in internal clinical standards and improved management of ARDS patients might account for systemic bias between groups. Finally, sample size is limited, hence solid conclusions on overall survival must be drawn with caution, especially within the ECMO subgroup.

## 6. Conclusions

The pathophysiology of early COVID-19 associated ARDS differs from Influenza-associated acute lung failure by sustained respiratory mechanics during the early phase of ventilation. Patients with early COVID-19 associated severe respiratory failure frequently exhibit less severely impaired lung mechanics. In these patients, the benefit from early intubation seems questionable, as long as it is not outweighed by the risk of patient self-inflicted lung injury.

## Figures and Tables

**Figure 1 jcm-11-06237-f001:**
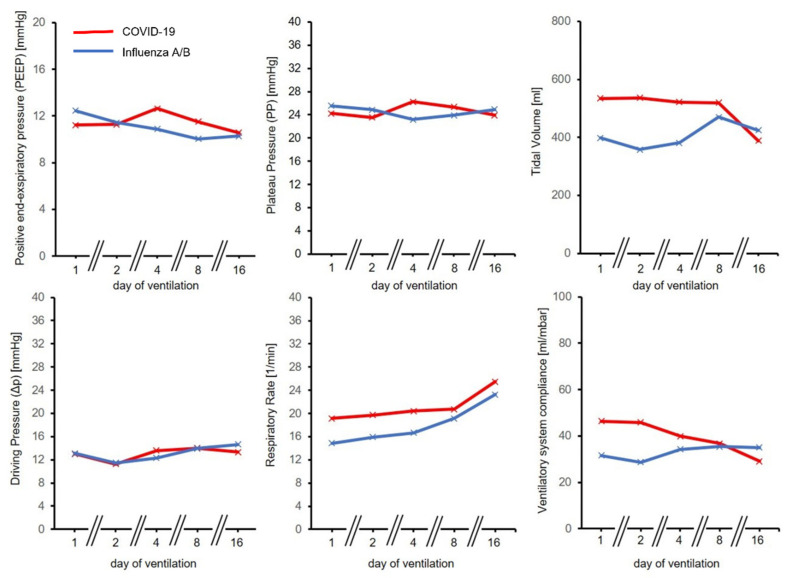
Development of ventilation parameters over time for COVID-19 and Influenza A/B. Time course was similar in both groups for positive end-expiratory pressure (PEEP), Plateau Pressure (PP) and Driving Pressure (ΔP). Respiratory Rate (RR), Tidal volumes and ventilatory system compliance were initially higher for the COVID-19 group.

**Figure 2 jcm-11-06237-f002:**
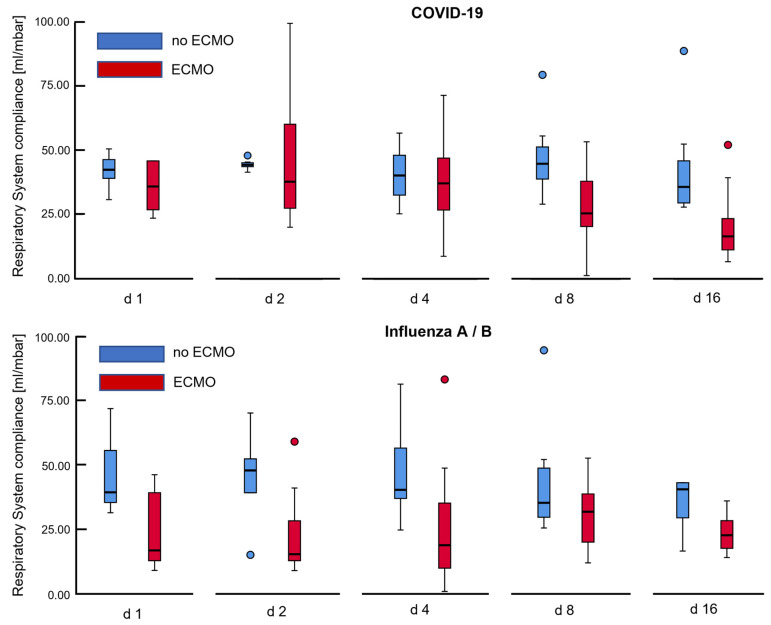
Development of respiratory system compliance over time for COVID-19 and Influenza associated ARDS. Histograms depict that ventilatory system compliance showed gradual decline over 2 weeks for COVID-19 patients on veno-venous extracorporeal membrane oxygenation (ECMO), while compliance was severely impaired in Influenza patients from the very beginning of invasive ventilation, improving insignificantly over time. The depicted box plots mark outliers with blue and red circles.

**Figure 3 jcm-11-06237-f003:**
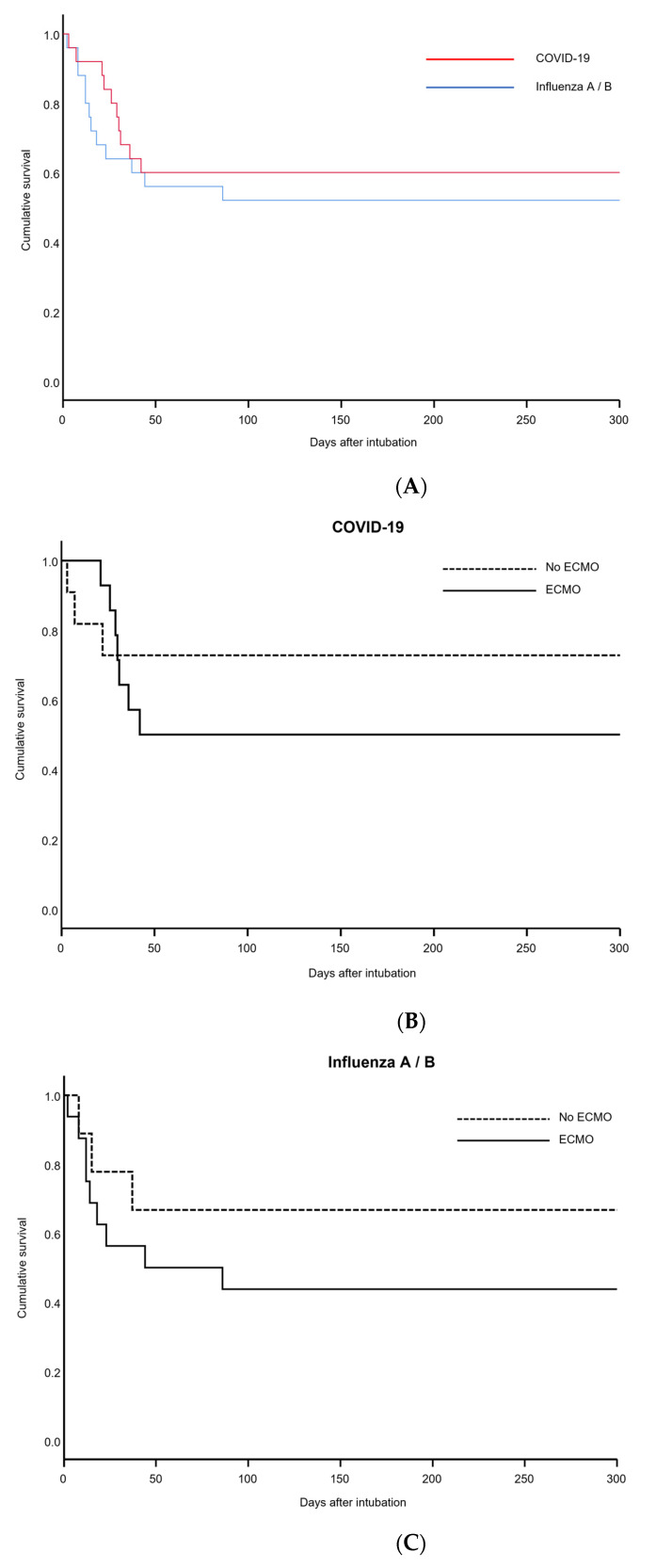
Kaplan–Meier plots for COVID-19 and Influenza patients. (**A**) Overall cumulative survival for COVID-19 and Influenza patients. (**B**) Subgroup analysis of cumulative survival for COVID-19 patients on ECMO and without ECMO. (**C**) Subgroup analysis of cumulative survival for Influenza patients on ECMO and without ECMO.

**Table 1 jcm-11-06237-t001:** Basic patient data.

	COVID (n = 25)	Influenza (n = 25)	*p*
Age on admission [years]	61.7 [53.1–68.1]	55.7 [45.5–65.4]	*p* = 0.082 *
Male/female	20/5	16/9	*p* = 0.210
Height [m]	1.77 [1.73–1.81]	1.73 [1.70–1.76]	*p* = 0.105
Weight [kg]	100.4 [91.6–109.2]	95.9 [81.3–110.5]	*p* = 0.573
Body-Mass Index (BMI) [kg/m^2^]	32.3 [29.0–35.5]	31.5 [27.0–36.0]	*p* = 0.781
Comorbidities:	Adiposity (19) Arterial Hypertension (13) Diabetes mellitus Type II (3) Coronary Heart Disease (3) Congestive Heart Failure (3) Past pulmonary embolism (3) Past stroke (4) Myasthenia gravis (1)	Adiposity (10) Arterial Hypertension (10) Diabetes mellitus Type II (9) Coronary Heart Disease (6) COPD (4) Chronic kidney disease (3) Epilepsy (2) Atrial fibrillation Cystic fibrosis Asthma	*p* = 0.010 ** *p* = 0.400 *p* = 0.047 ** *p* = 0.270

***** Mann–Whitney U test. ** Chi-squared test. COPD: Chronic obstructive Pulmonary Disease.

**Table 2 jcm-11-06237-t002:** Comparison of ventilation parameters on the first day of invasive ventilation.

	COVID-19	Influenza A/B	Significance
Arterial oxygen partial pressure [mmHg]	84.4 [76.8–92.1] n = 14	93.3 [71.8–114.8] n = 9	*p* = 0.313
Inspiratory oxygen fraction during non-invasive ventilation/HFNC	0.6 [0.49–0.70] n = 14]	0.75 [0.53–0.84] n = 9	*p* = 0.227
Arterial carbon dioxide partial pressure [mmHg]	42.1 [38.1–46.0] n = 14	45.0 [38.1–51.9] n = 9	*p* = 0.384
pH	7.39 [7.37–7.41] n = 14	7.34 [7.30–7.39] n = 8	*p* = 0.03
Horovitz-Index	127.9 [112.8–161.3] n = 14	135.4 [100.3–180.4] n = 9	*p* = 0.557
Positive end-expiratory pressure (PEEP) [cmH_2_O/mbar]	12.0 [9.6–13.5] n = 14	13.0 [9.0–15.5] n = 9	*p* = 0.369
Plateau pressure [cmH_2_O/mbar]	23.5 [20.8–29.0] n = 14	25.0 [24.0–28.5] n = 9	*p* = 0.561
Delta-P [cmH_2_O/mbar]	13.5 [10.8–16.0] n = 14	12.0 [11.0–14.5] n = 9	*p* = 0.643
Tidal volume [mL]	551 [434.5–593.8] n = 14	431 [210–572] n = 9	*p* = 0.124
Tidal volume per kg predicted body weight [mL/kg]	7.69 [7.12–8.12] n = 14	5.12 [3.14–8.40] n = 8	*p* = 0.059
Respiratory Rate * [min^−1^]	20.0 [15.5–21.3] n = 14	14.0 [12.5–17.5] n = 9	*p* = 0.011
Minute ventilation [L/min]	10.7 [7.2–12.2] n = 14	6.0 [2.5–10.1] n = 9	*p* = 0.013

***** The real frequency of assisted spontaneous breathing was counted whenever intended, and actual rates diverged by more than 2 per minute. HFNC: High-Flow Nasal Cannula.

**Table 3 jcm-11-06237-t003:** Comparison of ventilatory system compliance and respiratory ratio on the first day of ventilation.

	COVID-19	Influenza A/B	Significance
Ventilatory system compliance [mL/mbar]	40.7 [31.8–46.7]; n = 14	31.4 [13.7–42.8]; n = 9	*p* = 0.198
Ventilatory system compliance/kg idealized body weight [mL/mbar/kg]	0.57 [0.48–0.70]; n = 14	0.41 [0.20–0.59]; n = 8	*p* = 0.150
Ventilatory Ratio	1.57 [1.31–1.84]; n = 14	0.91 [0.44–1.38]; n = 7	*p* = 0.006

## Data Availability

Data can be provided on request addressed to the corresponding author. All data sharing statements are subject to conformity with German data protection legislation and rules (Datenschutzgrundverordnung-DGSVO).

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
