# Peer review of "Respiratory Physiology of COVID-19 and Influenza Associated Acute Respiratory Distress Syndrome"

_jcm, 2022, doi:10.3390/jcm11216237_

Round 1

Reviewer 1 Report

Thank you for your effort to present such interesting work.

I have some comments, I hope you can find them beneficial.

1- 1ry goal is not clear at the end of the introduction section.

2-I ask about any reference for guidelines of research you have mentioned on page 2 lines 99-100.

3- because of the significant difference between groups regarding gender, would you please add this point to the study limitations?

4- there is much duplication in data presentation e.g., page 4 lines 99-100 and table 1.

5- I ask about the p-value for gender and comorbidities.

6- regarding figure 1, please specify the colour coding for groups

7-Any further notes regarding studies stated as reference no 6and 21 on discussion page 9 line 292.

Reviewer 2 Report

Dear Authors,

My main comments are:

·       the study compared a population quite homogenous and included in 1 year, versus a not homogeneous historical population observed in 10 years;

·       I disagree with the conclusion about the suggestion of using awake ECMO for fighting the de-recruitment phenomenon. This because the recruitment is due to the combination of PEEP and Transpulmonary pressure applied to the patient lungs, particularly to the dependent zones. During spontaneous breathing without intubation the muscles could apply an inappropriate transpulmonary pressure and the caregiver cannot detect it, unless using an esophageal balloon for measuring the pleural pressure. This situation is facilitating the self-induced VILI, that it was one of the most important component of the high mortality rate observed in the COVID 19 patients, especially during the first wave.

·       The simple size is too small to asses as outcome the survival rate (particularly for the sub-group ECMO)

Minor comments:

The text need a language revision

The use of the italic characters it is not clear

Reviewer 3 Report

The manuscript by Kronibus et al. focuses on evaluating respiratory physiological parameters in covid versus influenza ARDS patients. Information from this study provided preliminary data suggesting compliance is higher in covid ARDS compared with influenza ARDS. The following is a synopsis of concerns raised with the data presented in the manuscript.

Major concerns:

1) How were the influenza ARDS control subjects recruited? Are they recruited retrospectively or prospectively?

2) Sample size may be too small to perform subgroup analysis regarding ECMO ouctomes.

3) Tidal volume received by Covid ARDS patients are higher than influenza ARDS patients.

Minor concern:

1) The format of references needs to be changed.

Round 2

Reviewer 2 Report

Dear Authors,

Thanks for your answers.

I am satisfied by the corrections.

Reviewer 3 Report

The authors have addressed my concerns.